# Experimental and Numerical Investigation on the Effect of Scratch Direction on Material Removal and Friction Characteristic in BK7 Scratching

**DOI:** 10.3390/ma13081842

**Published:** 2020-04-14

**Authors:** Wei Wang, Zhenping Wan, Shu Yang, Junyuan Feng, Liujie Dong, Longsheng Lu

**Affiliations:** School of Mechanical and Automotive Engineering, South China University of Technology, Guangzhou 510640, China; mewangwei@mail.scut.edu.cn (W.W.); zhpwan@scut.edu.cn (Z.W.); september_miracles@163.com (J.F.); dlj0777@163.com (L.D.); meluls@scut.edu.cn (L.L.)

**Keywords:** optical glass BK7, scratch direction, nanoscratching, material removal coefficient of friction

## Abstract

In order to study the influence of scratch direction on the deformation characteristics and material removal mechanism of optical glass BK7, nanoscratching experiments were conducted on a Nano indenter using Vickers indenter. Results indicate that the face-forward scratch is more likely to induce the initiation and propagation of lateral cracks, which is found to be more beneficial to material removal processes; in contrast, small chips and debris are released from the machined grooves without introducing lateral cracks in the edge-forward condition, leading to poor material removal efficiency. In addition, the choice of scratch direction can make differences to the elastic recovery rate of optical glass BK7. The results revealed that both the elastic recovery rate and the residual stresses of the material under the face-forward scratching are greater than those of the edge-forward scratching. A theoretical model for coefficient of friction (COF) under different scratch directions was established. It is found that the COF between indenter and workpiece in the edge-forward scratching is larger than the face-forward scratching under otherwise identical conditions, this finding is consistent with experimental results. A stress field analysis using finite element method (FEM) was conducted to understand the different crack initiation and propagation behaviors from different scratch directions. The current study discusses the significance of scratch direction on material removal behavior of optical glass BK7, and the results would encourage further research on investigating the connections between tool geometry and material removal mechanism.

## Highlights


Nanoscratching characteristics of optical glass BK7 using Vickers indenter under different scratch directions were experimentally investigated.Both the elastic recovery rate and surface deformation behavior of optical glass BK7 were greatly affected by the scratch direction.Lateral cracks were found to be more likely to initiate under face-forward scratch direction.A novel theoretical model incorporating the effect of scratch direction was developed to predict the coefficient of friction during scratching.Stress field analysis after scratching was conducted by finite element method to understand the different crack initiation and propagation behaviors from different scratch directions.


## 1. Introduction

Optical glass BK7 has been a promising material widely applied in aeronautics, laser technology, photoelectric communicational, and medical fields because of its stable mechanochemical properties and excellent optical uniformity [1,2,3,4]. For precision and high value-added applications, the material is required to provide accuracy and surface roughness at micro/nano scale. However, optical glass BK7 is a typical hard-brittle material which is difficult to machine for precision because of its high hardness and low fracture toughness [5]. Extensive studies investigating the material removal mechanism of hard-brittle materials have been conducted through scratching experiments at micro/nano scale, which provided in-depth understanding of the processing mechanism between abrasive grains and workpiece such as grinding and polishing.

Many efforts have been devoted to research the machining characteristics of brittle materials by single-point diamond scratching test. Researches have revealed that nanoscratching of hard-brittle materials is a complex process with multiple influencing factors, including scratch speed [6,7,8], shape and geometric parameters of tool [9,10], environment [11], and so on. In addition, many scholars have conducted studies to explore the effect of scratch directions in the processing of materials [12,13,14,15,16]. Yan investigated the material removal state under different scratch directions employing AFM on single crystal copper, it is found that the height of pile-up is greatly influenced by scratch directions [17]. Guo also found that the scratch depth is significantly affected by the normal load in different scratch directions of the tip-based micro/nano machining. Their findings suggested that three-dimensional micro/nano structures can be machined on the silicon base with the proper selection of the scratch direction. However, the scratch experiments using AFM probe-based nanoscratching method is limited to scratch depth to nanometer scale [18,19]. In addition to the scratching experiments by means of AFM probe tip, some scholars have also used Berkovich indenter for scratching investigation, but they rarely took scratch direction into account [20,21,22]. Zhang carried out the varied-cutting-depth nanoscratching experiments on 6H-SiC using the Berkovich indenter, in which the material removal behavior during scratching was found to be affected by scratch directions. The author also found that the ductile-removal mode takes place during the nanoscratching process [16]. Only a few studies have considered the effect of scratch direction on scratching characteristics [16], but the understanding is still limited.

To facilitate the understanding of nanoscratching process on hard-brittle materials, a comprehensive analysis of the stress field becomes important in providing insights into the fundamentals of material removal and surface characteristics [23,24,25,26]. Previous studies have indicated that the principle stresses and shear stress would rise with the increase of COF during processing [27]. However, the COF between the tool surface and workpiece is usually considered as constant in previous research which is different from the actual machining processes [28]. As a result, COF should be considered as a dominating factor that affects the deformation of the hard-brittle samples [29,30]. The aforementioned combination of multiple factors including the shape of tool and scratch direction combined together and collectively contributed to the stress field change, which eventually led to the deformation to different levels. Therefore, it is necessary to study the influence of scratch direction on the scratching characteristics of hard-brittle materials, including the elastic recovery rate of the material, the material removal mechanism, and the friction characteristics between the tool and the workpiece. In the current study, the elastic-plastic stress field analysis was conducted using finite element method (FEM) to support experimental results and provide guiding significance to experiments.

This paper aims to reveal the effect of scratch directions (face-forward and edge-forward) on the surface characteristics and material removal mechanism of optical glass BK7 material in the single-grit nanoscratch test. A theoretical COF model considering the effects of scratch direction, material elastic recovery rate, and geometry of the pyramid tip for nanoscratching was established and compared with the experimental results. The micro/nanoscale scratching characteristics of optical glass BK7 by using the quadrangular pyramid indenter were studied systematically. The effect of scratch direction on deformation characteristics and material removal behavior of optical glass BK7 was explained and verified through finite element analysis (FEA) by comparing with experimental results.

## 2. Experimental Details

### 2.1. Experimental Setup

In the present study, all scratching experiments were conducted on a G200 Nano Indenter (Keysight Technologies, Inc., Santa Rosa, USA) (shown in Figure 1) by Vickers indenter with average tip radius of 200 nm, face angle of 136°, and edge angle of 148°. By changing the orientation of the Vickers indenter as shown in Figure 2, the face-forward direction and edge-forward direction nanoscratching were realized, respectively. The material removal mechanism, as well as the initiation and propagation of microcracks of optical glass BK7 were investigated. The polished specimen was mounted on the workbench of G200 Nano Indenter for scratching experiments as shown in Figure 1. The nanoscratching experiments were conducted under constant load mode ranging from 10 mN to 50 mN (10 mN increment) as well as ramp load mode up to 50 mN with 100 um scratch length and 2 um/s scratch speed, the conditions were equivalent to the conditions of a quasi-static scratching. Each experimental set was repeated for five times with the same Vickers indenter to ensure repeatability. The experiments were all conducted under 24 °C room temperature and 60% relative humidity. As shown in Figure 2, all scratch tests were conducted with a quadrangular-based pyramid indenter with different directions (edge-forward and face-forward). For the purpose of illustration, the assumed elastic recoveries of the material under different scratch directions (edge-forward and face-forward) were highlighted in orange in Figure 2. The elastic recovery rate may significantly affect the actual contact area between the workpiece and the indenter. This is discussed in the later sections.

### 2.2. Specimen Characterization and Measurement

The specimen used in nanoscratching experiment was optical glass BK7 (7 mm × 7 mm × 3 mm) and the chemical composition is outlined in Table 1. All the samples were subjected to fine grinding before scratching to ensure the initial roughness (*R*a) being below 2 nm. The morphology of the specimen after edge-forward and face-forward scratching was observed by confocal laser scanning microscopy (Model: KEYENCE VK-X Series KEYENCE, Japan) and scanning electron microscopy (SEM, Merlin, Zeiss, Jena, Germany). The cross-sectional morphology, scratch depth, and residual depth of the scratch grooves were obtained by confocal laser scanning microscopy and atomic force microscopy (AFM, Dimension Icom, Bruker, Company, Gernamy).

## 3. Results and Discussion

Herein, the effects of scratch direction on elastic recovery rate, friction characteristics, surface deformation, material removal mechanism, and stress field distribution are discussed. Scratch experiments were conducted to verify the scratch simulation results by looking into the elastic recovery rate and friction characteristics under different scratch directions. The analysis procedure is shown in Figure 3.

### 3.1. The Effect of Scratch Direction on Elastic Recovery Rate

In the nanoscratch process of optical glass BK7, the scratch depth is different from the residual depth after scratching, a certain elastic recovery would occur [31]. The scratch elastic recovery has a significant influence on the material removal volume and the machining accuracy, and should be considered in the determination of scratching parameters [32]. In order to analyze the elastic recovery behavior of the optical glass BK7 under different scratch directions, we here define the ratio between the residual depth after the scratch process (residual depth) and the scratch depth during scratching as the scratch depth ratio [33].
(1)$$\lambda =\frac{h_{r}}{h_{s}}$$
where the scratch depth ratio of the material is λ, the residual depth is $h_{r}$, and the scratch depth is $h_{s}$. Thus, the elastic recovery rate η of the optical glass BK7 can be expressed as [34]:(2)η=(1−λ)×100%

As shown in Figure 2, the projected area of the interface for Vickers indenter tip at face-forward or edge-forward direction resulted in a difference between the hardness and elastic recovery rate of BK7. The variation between the scratch depth and the residual depth during scratching under different constant loads is shown in Figure 4. The error bars are two standard deviations in depth and they are estimated within 2%. It is shown that the scratch depth of the edge-forward direction was slightly larger than the scratch depth in face-forward direction. Under the same loading conditions, the residual depth from the edge-forward direction was also larger than the scratch residual depth in face-forward direction. Therefore, the two factors above should be incorporated into the analysis of the elastic recovery rate of BK7. By comparing the scratch depth ratio and the elastic recovery rate of the material, the obtained scratch depth ratio was around 0.413 (standard deviation is 0.0151) for face-forward direction and around 0.483 (standard deviation is 0.0165) for edge-forward direction. It should be noted that the smaller the residual depth, the bigger the residual stress after scratching and the higher the elastic recovery rate of the material [32]. Therefore, both the elastic recovery rate and the residual stress under the material under face-forward scratching were greater than those of the edge-forward condition. According to Figure 4 and Equation (2), the elastic recovery rate of BK7 in face-forward scratching was 58.7%, while it was 51.7% in edge-forward scratching, a 6% difference was observed.

### 3.2. The Effect of Scratch Direction on Friction Characteristics

Studies have shown that the friction characteristic in the scratching process is related to the stress state of the material, and the principle stress and shear stress in all directions increase as the COF increases [29]. It indicates that the feature of the COF in scratching is a major factor affecting the deformation of hard-brittle materials [35,36]. The change of COF also changes material deformation and mechanisms, which further affects the critical cutting depth of ductile to brittle transition for hard-brittle materials. Therefore, it is necessary to incorporate the friction characteristic between the tool and the workpiece into the current study. By referring to the traditional calculation method, the equivalent COF is composed of the ploughing COF and the interfacial COF [37,38,39]. In this paper, a theoretical model for calculating COF was developed to investigate the friction characteristic of optical glass BK7 in nanoscratching for both edge-forward and face-forward directions.

#### 3.2.1. Theoretical COF Model for Edge-Forward and Face-Forward Nanoscratching

In general, the friction force between indenter and groove surface is equal to the sum of the adhesion force and the ploughing force in the nanoscratching test [40,41], namely
(3)$$F_{t}=F_{A}+F_{P}$$
where $F_{A}$ and $F_{P}$ are the adhesion force and ploughing force, respectively. Furthermore, Williams [40] pointed out that the corresponding hardness value $H_{P}$ of the ploughing force $F_{P}$, the ratio of the ploughing force $F_{P}$ to the projected area $A_{t}$ along the scratch direction, is considered to be the energy used to replace the unit volume of material. According to Williams, the material resistance to penetration can be considered as a material constant, which is assumed to be the value of the ploughing hardness $H_{P}$, and is equal to the scratch hardness $H_{S}$. So the equation can be rewritten as [40]:(4)$$F_{t}=F_{A}+H_{P}\times A_{t}$$

Therefore, the overall COF can be expressed as:(5)$$\mu =\frac{F_{A}}{P}+\frac{H_{P}A_{t}}{H_{S}A_{n}}$$
where $A_{n}$ represents the projected area of the contact between the indenter and the material along the vertical direction, $A_{t}$ represents the projected area of the contact area between the indenter and the material along the scratch direction (the area highlighted in orange in Figure 2), and P is the normal load applied during the scratching process.

Now, assume $H_{S}$ = $H_{P}$ mentioned in the hypothesis, the overall COF becomes:(6)$$\mu =\mu_{A}+\frac{A_{t}}{A_{n}}$$

According to Figure 2b, it is evident that for edge-forward scratching, the values of $A_{t}$ and $A_{n}$ have a strong correlation with the indenter geometry because of the size effect and the elastic recovery rate of the material. Since the ploughing action plays a dominant role in ductile removal regime, the value of the ploughing part of the COF mainly depends on the plastic deformation in the scratch test. Thus, investigation was conducted to reveal the effect of face-forward and edge-forward directions on the COF under plastic deformation regime.

For face-forward scratching, the contact area between the Vickers indenter and the workpiece is shown in Figure 2a. The angle between the rake face and the flank face is 136°, i.e., the half tool apex angle is 68°. The half width of the rake face and the flank face of the indenter in contact with the workpiece are *b*_1_, *b*_2_ respectively, and they can be derived as:(7)$$b_{1}=h_{s}\tan\alpha$$
(8)$$b_{2}=(1-\lambda )h_{s}\tan\alpha$$
where α is the half apex angle of the indenter in face-forward scratching. Considering the elastic recovery of the material, the projection of the contact area between the indenter and the material along the vertical direction, namely $A_{n-ff}$, should be rewritten as:(9)$$A_{n-ff}=2b_{1}{}^{2}+b_{1}b_{2}+b_{2}{}^{2}=(4-3\lambda +\lambda^{2})h_{s}^{2}\tan^{2}\alpha$$

Similarly, the projection of the contact area between the indenter and the material along the scratch direction, namely $A_{t-ff}$, should be rewritten as:(10)$$A_{t-ff}=h_{s}^{2}\tan\alpha$$

Then, the ploughing COF in face-forward scratching can be derived as:(11)$$\mu_{p-ff}=\frac{A_{t-ff}}{A_{n-ff}}=\frac{1}{(4-3\lambda +\lambda^{2})\tan\alpha }$$

As for the edge-forward scratching, the projections of the contact area in vertical direction $A_{n-ef}$ and in scratch direction $A_{t-ef}$ should be rewritten as:(12)$$A_{n-ef}=(b_{1}+b_{2})b_{1}=(2-\lambda )h_{s}^{2}\tan^{2}\beta$$
(13)$$A_{t-ef}=h_{s}^{2}\tan\beta$$
where *β* is the half apex angle of the indenter in edge-forward scratching. Thus, the ploughing COF in edge-forward scratching can be expressed as:(14)$$\mu_{p-ef}=\frac{A_{t-ef}}{A_{n-ef}}=\frac{1}{\left(2-\lambda \right)\tan\beta }$$

#### 3.2.2. Comparison of the Theoretical and Experimental Results

In addition to the constant loading conditions, an exploratory study was conducted on the nano indenter utilizing the lateral force module (LFM) function. In the LFM function, the applied load (in the normal direction) varied following a ramp path, and the tangential force was measured by the equipment and reported after the test. Thus, the COFs of face-forward and edge-forward directions under ramp loading conditions can be obtained. Figure 5a shows the normal force, tangential force, and COF of the Vickers indenter from different scratch directions under ramp loading condition and Figure 5b shows the COF of the Vickers indenter from different scratch directions under constant normal loading condition.

It is evident that the tangential force of the edge-forward scratching is greater than that of the face-forward condition; more specifically, the COF of the edge-forward scratching is greater than that of the face-forward condition. As mentioned above in Equations (11) and (14), assuming the same elastic recovery rate of the material, the COF under the edge-forward scratching is larger than that of the face-forward, which is consistent with the experimental results.

As noted by Gu et al. [42], no matter the test was under constant load or varying scratch depth, the average scratch depth ratio of BK7 under certain load (P < 130 mN) was 0.359 with a standard deviation of 0.0045. Taking λ = 0.359 into Equations (11) and (14) above, the ploughing term of the overall COF under face-forward scratching became 0.132 and 0.152 under edge-forward scratching. Substituting scratch depth ratios λ_ff_ = 0.413 and λ_ef_ = 0.483 from Section 3.1 into Equations (11) and (14), the overall COF became μ_ff_ = 0.138 and μ_ef_ = 0.189 respectively. It should be noted that the theoretical value was smaller than the experimental one since the influence of the COF adhesion term was not taken into consideration. As for the adhesion term μ_A_, it can be expressed as $\mu_{A}\propto \left(2/\pi \right)(s/\sqrt[{3}]{{\left(v/h\right)}^{4m}/E})$, where s is the shear strength of the interfacial, v is scratch velocity, h is scratch depth [43]. It can be obtained that μ_A_ is related to the scratching speed, and can be assumed to be a constant under different scratch directions. To a certain extent, the theoretical model reveals that the coefficient of friction is greater under edge-forward scratching.

### 3.3. The Effect of Scratch Direction on Surface Deformation, Lateral Cracks Development, and Material Removal Behavior

The specimen was ultrasonically cleaned in ethanol-acetone solution for ten minutes after the scratch test. The morphologies of the scratch groove were observed by confocal laser scanning microscopy and SEM. The results of face-forward and edge-forward scratching with different scratch depths were discussed in the following sections.

#### 3.3.1. Surface Deformation and Material Removal Behavior in Face-Forward Scratching

Optical micrographs of BK7 samples (face-forward scratching) from different scratch depths are shown in Figure 6 (*h* stands for scratch depth here). The results indicate that the initiation of lateral cracks during scratching has a strong dependence on the scratch direction. It can be clearly observed that bright-flaky regions were formed on both sides of the groove, which were lateral cracks nucleating near the plastic deformation zone and expanding laterally on a plane parallel to the specimen surface. The onset position of lateral cracks was identified by observing the groove under confocal laser scanning microscopy. As the scratch depth increased, the lateral cracks further propagated. The initiation depth of lateral crack was determined by the brightness of the spot in optical micrographs. In this study, the damage zone size, which is defined as the average width of the bright areas on both sides of the scratch groove, increased as the scratch depth increased (Figure 6). Therefore, the material removal volume during precision and ultra-precision processing can be better evaluated and estimated based on the learning between the damage zone size and the scratch depth (i.e., 0~900 nm) from single grit scratching.

When BK7 was scratched at a small depth (i.e., 0–500 nm), the force was insufficient to cause lateral cracks to propagate in the specimen surface. Therefore, no obvious chipping was observed around the scratch groove. When the scratch depth was deep (i.e., *h* > 500 nm), it can be observed that lateral cracks propagated upwards to the specimen surface, and eventually led to material removal. It should be noted that at smaller scratch depths, only lateral crack is observed. It can be seen from Figure 7 that when the scratch depth increases, radial cracks can be observed on the scratched. surface. Meanwhile, the material removal process results not only from the expansion of lateral cracks, but also from the interaction between lateral and radial cracks at this stage.

#### 3.3.2. Surface Deformation and Material Removal Behavior in Edge-Forward Scratching

The surface morphologies of edge-forward scratching under different scratch depths from 200 nm to 1 μm are illustrated in Figure 8, which were measured by AFM. It is evident that, in the edge-forward scratching process, continuous lateral cracks were not observed and chips were easily generated at both ends of the scratched groove. Moreover, the amount and morphologies of the chips varied with varying scratch depths. When BK7 was scratched by edge-forward direction at a small depth of 200 nm, there were hardly any cracks and burrs on the scratched surface. As shown in Figure 8a, material built up on both sides of scratched groove and plastic flow appeared. It is worth noting that the angle between the plastic flow lines and scratch direction was approximately 42° which showed a certain regularity. This angle was approximately equal to the angle between the edge of the Vickers indenter and scratch direction under edge-forward scratching. When the scratch depth was 400 nm as shown in Figure 8b, not only plastic flow, but also typical ductile-removal including scattered chips and strip chips occurred on the scratched surface. Moreover, from Figure 8b–f, it can be observed that as the scratch depth increased, the contact between the indenter and the material became more severe, and an increased amount of scattered chips were created on both sides of the scratched groove owing to the shearing flow. The chips were evenly distributed and the length of the strip chips were longer. When the scratch depth increased up to 1 μm, the length of the strip chip was about 3.95 μm.

As shown in Figure 8c, when BK7 was scratched up to 500 nm, serrated burrs were found on the side of the scratched groove. Furthermore, scattered chips agglomerated into block chips. With the increase of the scratch depth as show in Figure 8c–f, the formation of the sharp angle of the burrs became much more conspicuous and was densely distributed at the edge of the scratch grooves with larger area, especially when the scratch depth reached 800 nm to 1 um. As shown in Figure 8e,f, the generated chips were continuous, elongated, and curled, which were mainly observed on the edge side of the scratch groove without separation. In addition, some small chips aggregated together and formed into block chips because of their high-surface-energies as shown in Figure 8f. During the edge-forward scratching process, friction force grew up because of the fact that the extrusion between the diamond indenter and the specimen would enlarge along the direction of perpendicular scratching with the increase of scratch depth. Furthermore, greater friction force would result in an increasing number of chips, which was more likely to cause chip breakage.

The results indicate that the difference in scratch direction has a significant influence on the surface deformation characteristic and material removal mode at the same scratch depth. Under the same scratching conditions (including scratch depth, scratch speed, and scratch length), lateral crack-induced subsurface damage was not observed in edge-forward scratching, and the chips were mostly discontinuous and small in size which can be expelled from the scratch groove easily. In contrast, face-forward scratching was more prone to the initiation and continuous propagation of lateral cracks than edge-forward scratching. With the increase of scratch depth, the lateral cracks propagated forward and bulged to the sides of the scratched groove more obviously, leading to more material removal.

### 3.4. Numerical Simulation by FEM

In micro/nano scratching of hard-brittle materials, the surface and subsurface morphologies of the specimen are closely related to its stress state. In the following section, FEM was utilized to study the stress distribution of BK7 during single grit scratching. The relationship between the stress distribution and the initiation of the lateral crack in different direction scratching was analyzed, which could provide further explanation and certain guiding significance to the experimental results. The model mainly investigated the influence of the scratch direction on the sequence and initiation of lateral cracks, without considering the position and propagation of cracks during scratching process. Therefore, the isotropical bilinear elastic-plastic constitutive equation for BK7 was adopted to investigate the relationship between the stress field change and the lateral cracks initiation under different scratching conditions.

In order to obtain the stress state under different scratch depths, the simulation used a gradually increasing depth method and the parameters related to scratching simulation are shown in Figure 9. The Vickers indenter was assumed to be a rigid body, the geometric parameters of the indenter, the scratching length, and other conditions were kept unchanged. Given the very low scratch speed, a quasi-static approach was used in the simulation. The tool-workpiece engagement is shown in Figure 10a, and the cross sectional view in simulation is shown in Figure 10b,c, with the X direction being perpendicular to the scratch direction, the Y direction being the depth of the scratch along the normal-load direction, and the Z direction being along the scratch direction. To ensure the accuracy and efficiency of this simulation, the seeds of the edges in the X axis were set to be much dense when it approached the center, with a minimum mesh size of 50 nm at the center, as shown in Figure 10b,c.

According to the literature survey on the hard-brittle materials scratching, it is believed that the sequence of crack generation is: median crack, lateral crack, and radial crack [44]. The initiation and development of different cracks depend on the stress state. Previous studies illustrated that principle stress $\sigma_{xx}$ is responsible for median cracking, $\sigma_{yy}$ for lateral cracking, and $\sigma_{zz}$ for radial cracking [29,45,46]. In the highly nonlinear analysis of commercial software ABAQUS, the normal stress S_11_ along the scratch direction, the normal stress S_22_ in the vertical direction, and the normal stress S_33_ perpendicular to the scratch direction are the main driving forces for the initiation of the median crack, the lateral crack, and the radial crack, respectively.

According to the principle of Weibull’s fracture stress distribution, the Weibull fracture probability of the *i*th Gaussian point in the model can be expressed as [36]:(15)$$G_{i}=1-\exp\left\{-V_{i}{\left(\frac{\sigma_{1i}}{\sigma_{0}}\right)}^{t}\right\}$$
where $V_{i}$ is the volume of the *i*th Gaussian point, $\sigma_{1i}$ is the maximum normal stress, $\sigma_{0}$ and *t* are the Weibull constant of the material. It can be obtained from Equation (15) that the greater the probability of brittle fracture in the region, the larger the principle normal stress. Therefore, the initiation of lateral crack can be predicted by analyzing the maximum normal stress in the vertical direction during the scratching process.

To analyze the effect of normal stress S_22_ on the lateral crack development during face-forward and edge-forward scratching, the stress field diagrams after unloading and the maximum S_22_ value at the cross-section Z = 40 um (as shown in A-A section of Figure 9.), Z = 60 um (as shown in B-B section of Figure 9.) corresponding to the scratch depths of 200 nm and 300 nm are shown in Figure 11 and Figure 12. It can be found that stress S_22_ was zero when the indenter did not reached the scratching section, no matter what the scratching direction is. When the indenter passed through the studied cross section, the stress S_22_ suddenly increased, and the normal stress of the vertical section increased accordingly as the scratch depth increased. This suggests that the lateral crack initiation was affected by the residual stress field after scratching. Moreover, when the residual tensile stress became greater than the tensile strength of the material, the lateral crack initiated and propagated. Furthermore, it is found that for the same cross section of the specimen, the scratch directions (face-forward and edge-forward) had different effects on the normal stress in the vertical direction. In particular, the normal stress in the vertical direction of edge-forward scratching is smaller than that of the face-forward. It suggests that the probability of lateral crack initiation was far lower in edge-forward scratching. In addition, the lateral cracks were more prone to be initiated in face-forward scratching, which would lead to material removal during scratching at the same time. We can conclude that experimental results coincide with simulations.

## 4. Conclusions

In the current study, nanoscratching experiments were conducted on optical glass BK7, using a quadrangular pyramid probe-based indenter to investigate the influence of the indenter direction (face-forward and edge-forward) on the material removal mechanism and the material deformation characteristics including the elastic recovery rate and the COF. The morphology and initiation of the lateral cracks in face-forward and edge-forward scratching were investigated by experiments and FEM simulations. Furthermore, considering the stress distribution of the workpiece, the relationship between COF and scratch direction was investigated theoretically and numerically. By studying the surface deformation and material removal mechanism of optical glass BK7 under different scratch directions via single grit scratching, we can better understand the influence of abrasive grain arrangement on the quality of the workpiece during grinding process. Based on the results, the following conclusions can be drawn:(1)The results showed that both the elastic recovery rate and the residual stress of the material under the face-forward scratching were greater than that in the edge-forward scratching.(2)Scratch directions have a significant influence on the lateral crack generation and the material removal of optical glass BK7. It is found that face-forward scratching was more prone to the initiation and continuous propagation of lateral cracks than edge-forward scratching, which would eventually lead to more material removal under the same scratching condition, this is consistent with the results of the FEM simulation.(3)A theoretical model for COF incorporating the scratch direction effect was established and discussed. A more systematic nanoscratching COF model for Vickers indenter was established. The influences of the indenter including angle and the scratch direction were considered in the developed theoretical model and discussed analytically and experimentally. The results showed that COF in face-forward scratching was smaller than the edge-forward scratching.(4)The scratch direction based on edge-forward or face-forward in this study can be appropriately selected according to the morphology and surface quality of the machined groove. The face-forward scratch is more likely to introduce the initiation and propagation of lateral cracks to the surface because of the larger residual stress, while the edge-forward scratch is more likely to cause the chip to discharge from both sides of the groove because of the larger COF. The experimental results matched the theoretical COF model and FEM simulation well. This is considered to be more beneficial to material removal.

## Figures and Tables

**Figure 1 materials-13-01842-f001:**
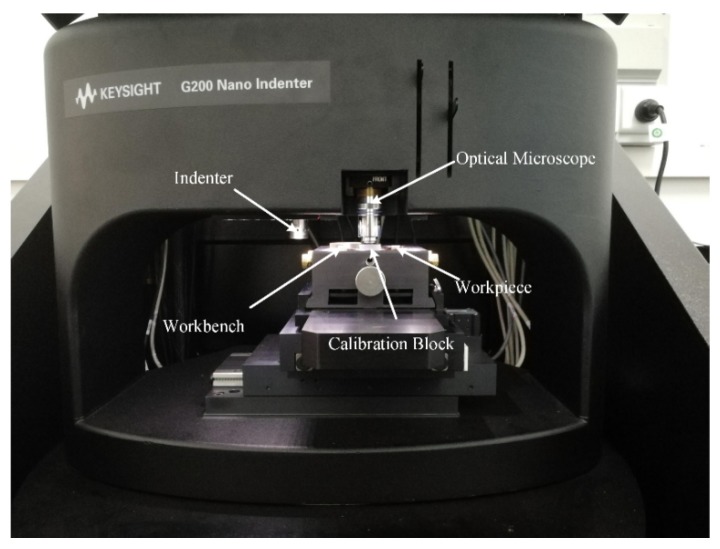
Experimental setup.

**Figure 2 materials-13-01842-f002:**
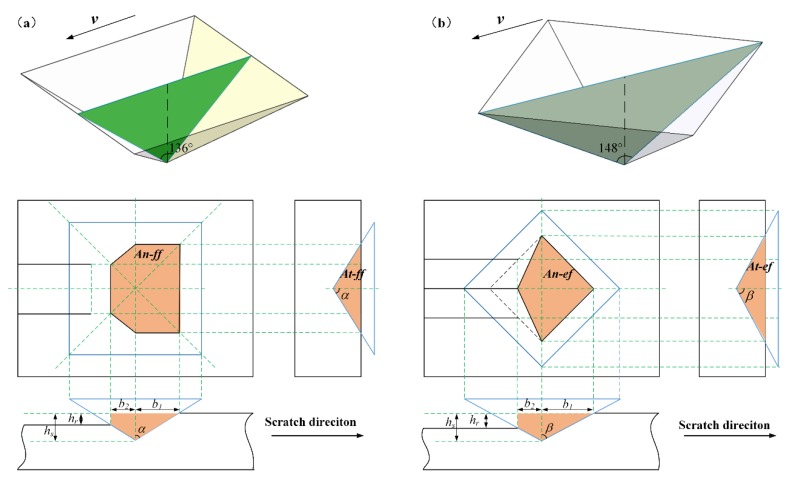
Projected areas of the interface for Vickers indenter tip with respect to different scratch directions: (**a**) face-forward; (**b**) edge-forward.

**Figure 3 materials-13-01842-f003:**
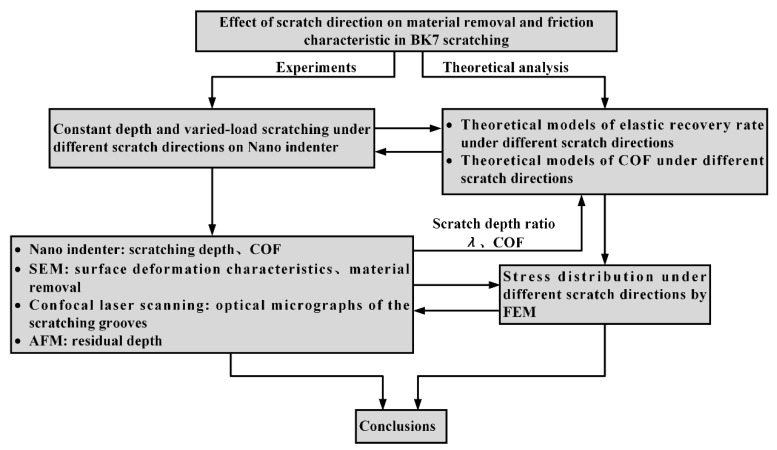
The schematic flow chart for study the effects of scratch directions.

**Figure 4 materials-13-01842-f004:**
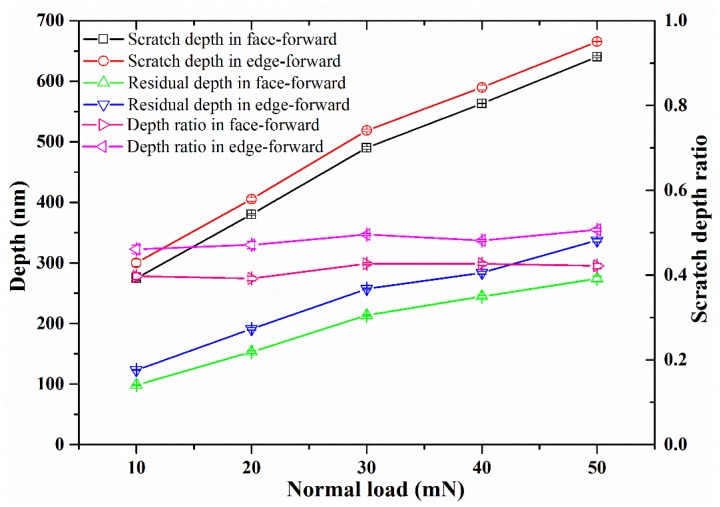
Scratch depth, residual depth, and depth ratio curve under different scratch directions.

**Figure 5 materials-13-01842-f005:**
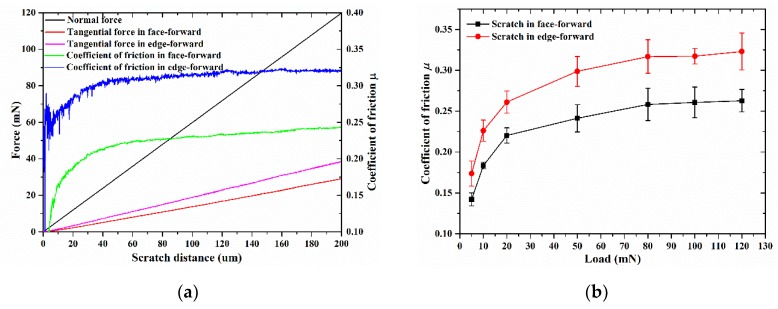
The experimental coefficient of friction (COF) under different scratch directions. (**a**) Normal force, tangential force and COF under ramp loading condition; (**b**) relationship between the normal load and COF under edge and face-forward directions.

**Figure 6 materials-13-01842-f006:**
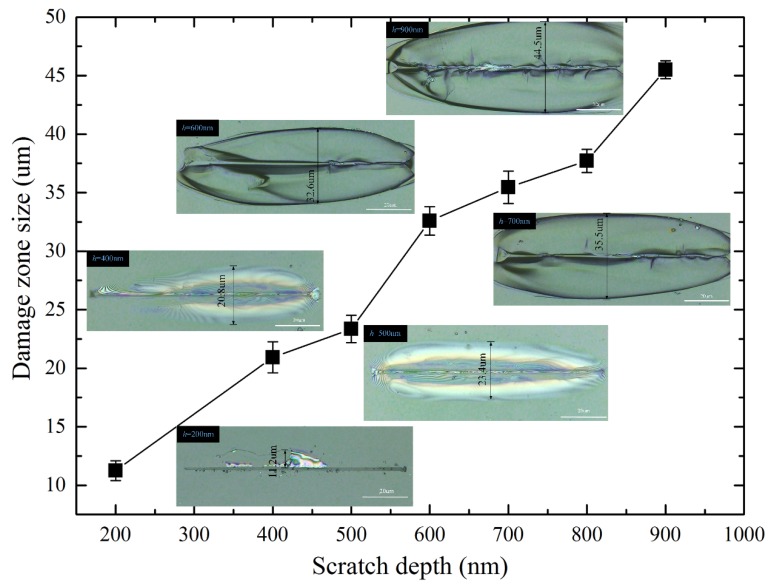
Resulting damage zone size due to lateral crack propagation versus scratch depth (imbedded images are optical micrographs of the scratching grooves in face-forward scratching under different scratch depths).

**Figure 7 materials-13-01842-f007:**
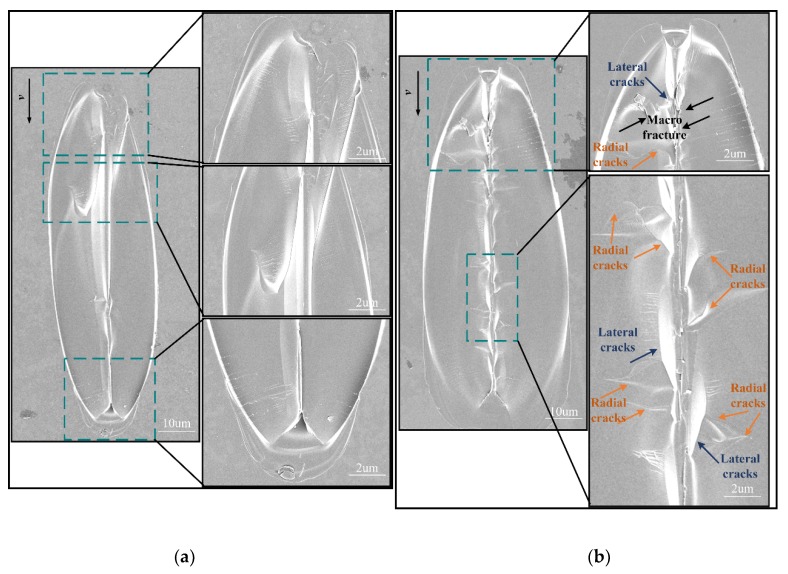
Images of scratch grooves in face-forward scratching: (**a**) 600 nm scratch depth; (**b**) 900 nm scratch depth.

**Figure 8 materials-13-01842-f008:**
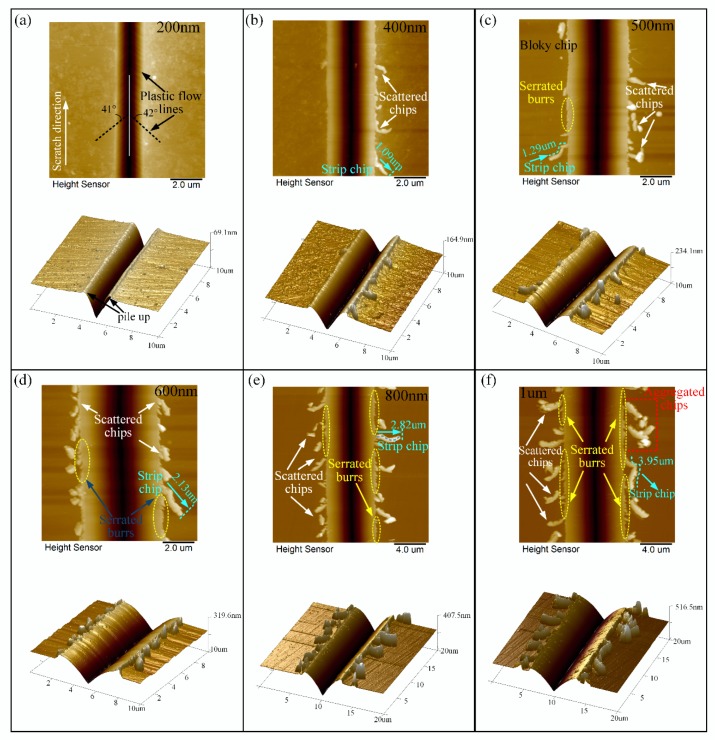
Surface deformation characteristic of edge-forward scratching under different scratch depths: (**a**) 200 nm; (**b**) 400 nm; (**c**) 500 nm; (**d**) 600 nm; (**e**) 800 nm; (**f**) 1 μm.

**Figure 9 materials-13-01842-f009:**
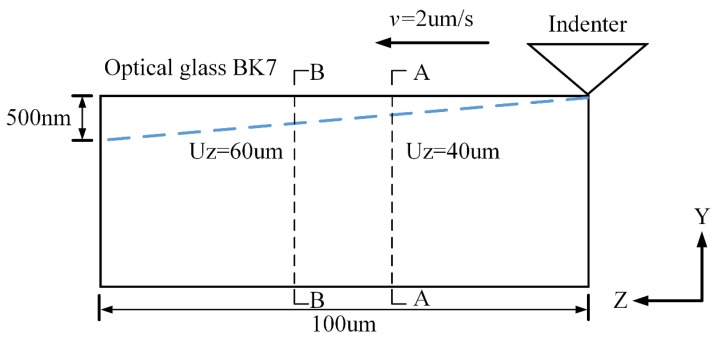
Illustration of a gradually increasing depth scratching in finite element method (FEM).

**Figure 10 materials-13-01842-f010:**
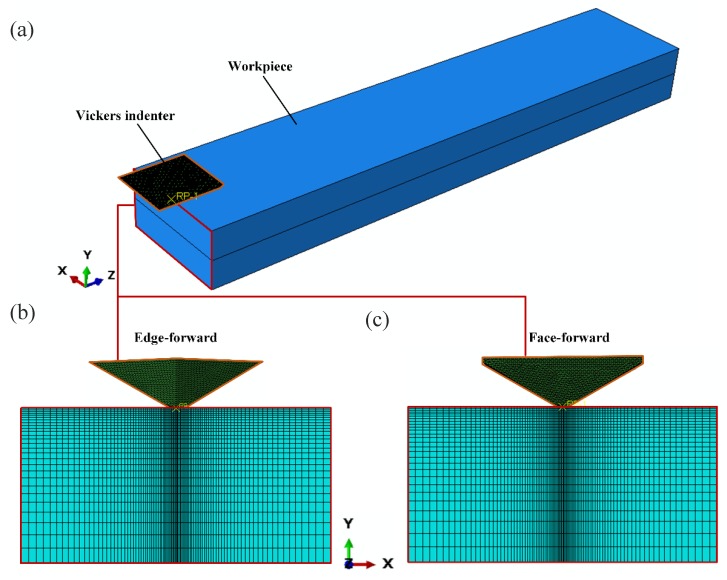
The three-dimensional finite element model. (**a**) the three-dimensional FEM; (**b**) the side view of edge-forward; (**c**) the side view of face-forward.

**Figure 11 materials-13-01842-f011:**
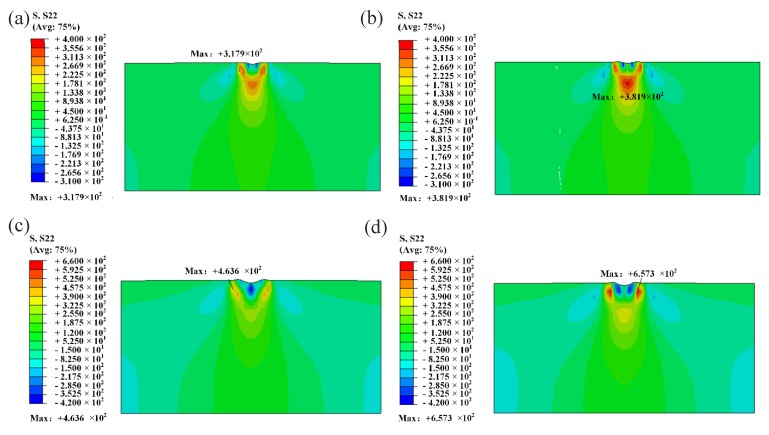
Cross-sectional stress field after unloading under different scratch directions. (**a**) cross-section at Z = 40 um under edge-forward; (**b**) cross-section at Z = 40 um under face-forward; (**c**) cross-section at Z = 60 um under edge-forward; (**d**) cross-section at Z = 60 um under face-forward.

**Figure 12 materials-13-01842-f012:**
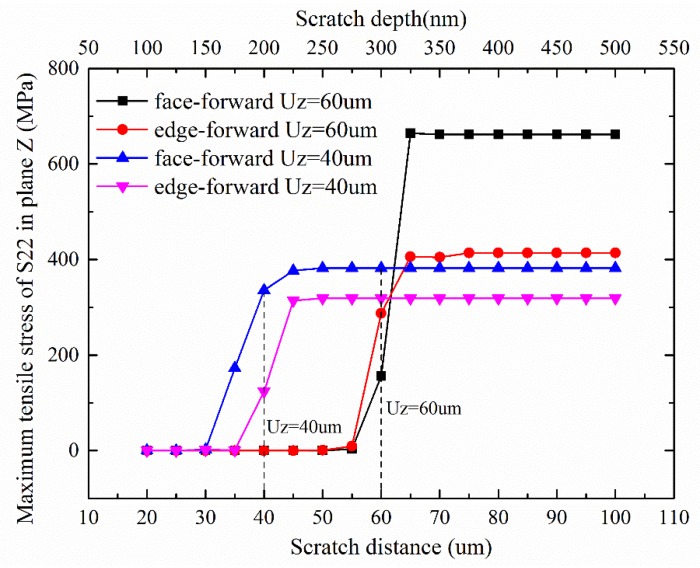
The maximum value of the cross section S_22_ stress with the scratch depth under different scratch directions.

**Table 1 materials-13-01842-t001:** Composition of the optical glass BK7 used in the experiments.

Material	Chemical Composition (wt %)					
	SiO_2_	B_2_O_3_	K_2_O	BaO	Na_2_O	As_2_O_3_
BK7 Glass	69.13	10.75	6.29	3.07	10.40	0.36

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
