# Peer review of "Experimental and Numerical Investigation on the Effect of Scratch Direction on Material Removal and Friction Characteristic in BK7 Scratching"

_materials, 2020, doi:10.3390/ma13081842_

Round 1
Reviewer 1 Report
This paper described the effect of scratch direction on the material removal and friction behavior of optical glass BK7. It is interesting to use a model considering the elastic recovery, but however, there are some points I wondered, and additional explanations are needed.
1) p.2, line 63:
The first author of the reference [16] is Dr. Zhang based on your reference list. Please confirm it.
2) Fig. 2:
What did the orange areas in the side view signify?
3) Equation (9):
The An-ff is (4-3λ+λ^2)*(hs^2)*(tanα^2), isn’t it? Is that an error in calculation?
4) Equation (10):
The projected area along the scratch direction must not be affected by the elastic recovery rate λ, only affected by the scratch (indentation) depth hs. In fact, the At-ff should be (hs^2)*(tanα). Please re-calculate the equation.
5) Equation (11):
If Eqs. (9) and (10) are wrong as aforementioned, Eq. (11) will be completely different. Eqs. (13) and (14) have the same issues as well.
6) p. 8, line 243:
When taking λ=0.359 into Eqs. (11) and (14), the resultant values were 0.089 and 0.090. That should be due to an error in calculation or wrong equation. There are the same situations for λ=0.413 and 0.483.
7) p.13, line 351:
You don’t have Fig. 10 (d).
8) Fig. 12:
You claimed that the lateral cracks were not observed in the edge-forward scratching unlike the face-forward scratching, and also that the lateral cracks initiated when the residual tensile stress exceeded the tensile strength. According to those, a certain criterion such as tensile strength should exist between the tensile stress for edge- and face-forward scratching in Fig. 12. Did you compare the stress in Fig. 12 with the tensile strength? Otherwise, the paper is structured illogically.
9) Conclusions (4):
You concluded that the experimental results matched the theoretical COF model. However, you explained that the theoretical COF was smaller than the experimental one since the effect of adhesion term was not taken into consideration and it was not investigated in the current study. From that, you cannot say that the experimental results matched the theoretical COF model. Furthermore, the relationship between the friction behavior and the material removal was unclear. In other words, the section of friction behavior is considerably independent from the other section. The reviewer could not catch the aim to investigate the friction behavior.
Reviewer 2 Report
The problem statement needs to be rewritten.
The main contribution of this work isn't clear.
The industrial application of such approach needs to be clearly presented.
More recent references should be added.
Round 2
Reviewer 1 Report
It seems to be acceptable in present form.